# The Dynamics of Pheromone Release in Two Passive Dispensers Commonly Used for Mating Disruption in the Control of *Lobesia botrana* and *Eupoecilia ambiguella* in Vineyards

**DOI:** 10.3390/insects15120962

**Published:** 2024-12-03

**Authors:** Marta Corbetta, Luca Bricchi, Vittorio Rossi, Giorgia Fedele

**Affiliations:** 1Department of Sustainable Crop Production (DiProVeS), Università Cattolica del Sacro Cuore di Piacenza, 29122 Piacenza, Italy; marta.corbetta2@unicatt.it (M.C.); vittorio.rossi@unicatt.it (V.R.); 2Consorzio agrario Terrepadane Scrl, Via Colombo 35, 29122 Piacenza, Italy; l.bricchi@terrepadane.it

**Keywords:** *Vitis vinifera*, integrated pest management, sex pheromones, environmental conditions, mathematical model, tortricid moths

## Abstract

Pheromone release is very important for mating disruption (MD) efficacy, and environmental conditions must be considered for their effect on pheromone release (PR). The aim of this study was to evaluate the effect of weather conditions on the pheromone release patterns of two passive dispensers used for the MD of grapevine moths. The study was conducted in four vineyards in northern Italy. For both dispensers, equations based on the accumulated temperature and vapor pressure deficit explained the PR dynamics with high accuracy. These equations can provide information on the PR during the grapevine-growing season for each dispenser type and may support a better management of MD against moths. These equations should also be linked to mathematical models predicting the phenology of grapevine moths populations to inform farmers about pheromone emission at the time of adult flights. However, the equations should be further validated against independent data collected under diverse environmental conditions before use.

## 1. Introduction

Two tortricid moths, the European grapevine moth (EGVM) *Lobesia botrana* (Denis & Schiffermüller) (Lepidoptera: Tortricidae) and the European grapeberry moth (EGBM) *Eupoecilia ambiguella* (Hübner) (Lepidoptera: Tortricidae), are the most important arthropod pests of grapevine (*Vitis vinifera* L.) in central Europe and Mediterranean areas [1,2,3,4]. These species have a considerable economic impact on grape yield and fruit quality, causing damage directly by larval feeding on grape flowers and berries and indirectly by increasing the susceptibility of berries to molds caused by *Botrytis cinerea* and other fungi [5]. The extent of the damage depends on the number of pest generations and a range of cultivar-specific factors, agronomic practices, and weather conditions [6,7].

Insecticides used against the two moths have gradually been replaced in integrated pest management (IPM) programs by more selective and less hazardous compounds, such as neurotoxic insecticides (e.g., spinosyns), chitin synthesis inhibitors, molting accelerating compounds, microbial insecticides, and some avermectins and anthranilic diamides [8]. Synthetic pheromones (i.e., behavior-modifying chemicals) are also widely used because of their considerable potential in IPM [9,10].

Mating disruption (MD) is the most effective and used pheromone-based technique for controlling grape moths in European wine-growing areas [6,8]. MD is an environmentally friendly pest control method that allows growers to significantly reduce pesticide use due to its high effectiveness, specificity, and lack of toxic residues [11,12].

The process at the base of MD (i.e., the disturbance of a male seeking a female) is the result of physiological and behavioral effects given by the emission into the vineyard environment of synthetic female sex pheromones by specific dispensers [13,14,15,16]. Also, females respond to their conspecific sex pheromone; for instance, MD affects the flight of females in *L. botrana*, causing short bouts of flying, increasing the age of mating, and reducing fecundity with a decrease in the amount of ovipositions [12]. Overall, a successful MD is indicated by a decrease in male catches during peak target flights, a reduction in insect mating, and a lower incidence of pests or damage compared to untreated areas [17].

Hand-applied dispensers (i.e., twist tie ropes, twin ampoules, and membranes) consist of small containers of plastic materials filled or impregnated with pheromones released passively through their walls [11]. The active ingredient in these dispensers is the pheromone component (*E*,*Z*)-7,9-dodecadienyl acetate, which is released into the air at a rate of 50–60 µg/h per dispenser [18]. These dispensers are evenly distributed in the vineyard (250–500 units on average per hectare) and attached to wires or vine shoots at the grape cluster layer with the leaves protecting them from direct sunlight and high temperatures [8].

Apart from the active ingredient, there is a large variation in the load, as well as in the physical and chemical features of the dispensers, including the size, shape, and thickness of the dispenser walls, leading to differences in release rates and dispenser lifespan [6]. The regular release of adequate levels of pheromone, good protection of the active ingredient from degradation, effective cover of the full seasonal activity of the target pests, and affordable costs for growers are prerequisites for reliable and efficient commercial MD dispensers [10,19]. Pheromone release is so important for MD efficacy that the pheromone remaining in the dispensers during the season can be considered an indicator of dispenser efficiency over the season [20].

Despite intensive research and substantial experience gained during the past few decades, many factors have limited the number of hectares of vineyards under pheromone-mediated MD against grapevine moths. These factors may affect pheromone concentration, homogeneity, and atmospheric distribution in vineyards. These factors are strongly influenced by plant spacing, training system, plant canopy, and leaf density according to the vine phenological stage because of the ability of foliage to absorb the pheromone [6]. A fully developed canopy in summer retains a higher concentration of pheromone than sparse vegetation in spring, and the pheromone concentration takes longer to decline within full vegetation than a growing canopy [21].

Environmental conditions, such as temperature, relative humidity, wind, and rainfall, may also affect the pheromone release rate [22]. Most of the studies concerning the effect of weather variables on pheromone release were conducted under environment-controlled conditions, usually by testing single variables (e.g., different, constant temperature levels, as in Hofmeyr and Burge [23]) or, less frequently, some combinations of two variables (e.g., temperature and relative humidity, as in Zhu et al. [24]). Under field conditions, however, the weather variables fluctuate, may manifest peaks not considered in laboratory experiments, and affect pheromone release altogether. Studies under natural conditions, however, are still limited in number [25].

We aimed to study the effect of weather conditions on pheromone release patterns under field conditions. We correlated the pheromone release (PR) by two types of MD passive dispensers largely used against EGVM and EGBM with different weather variables directly measured by weather stations or calculated from the measured data. Specifically, we calculated a temperature-dependent pheromone release rate (PRR) and vapor pressure deficit. The temperature-dependent PRR was calculated through an equation developed from a preliminary experiment conducted under controlled conditions.

## 2. Materials and Methods

### 2.1. Pheromone Dispenser Types

Two MD pheromone dispensers were compared: (i) Isonet^®^ L PLUS (Biogard, CBC Europe S.r.l., Grassobbio (BG), Italy; hereinafter referred to as Isonet), which contains (*E*,*Z*)-7,9-dodecadienyl acetate and (*Z*)-9-dodecenyl acetate (213 mg/dispenser); and (ii) Rak^®^ 2 Max (BASF Agricultural Solutions Italia, Cesano Maderno (MB), Italy; hereinafter referred to as Rak) which contains (*E*,*Z*)-7,9-dodecadienyl acetate and *N*-dodecenyl acetate (361 mg/dispenser). Isonet consists of two parallel microcapillaries (20 cm length) welded at the ends of plastic polymer, one containing the aluminum wire that allows its application and the other filled with the pheromone. The Rak dispenser comprises two ampoules in inert plastic material (3 × 8 cm), and an inverted U-shape hook is used to hang the dispenser onto the wires in the vineyard.

### 2.2. Effect of Temperature on Pheromone Release

Dispensers were placed in incubators at six constant temperature regimes (5, 10, 15, 20, 25, and 30 °C). Five dispensers were used per type and temperature. The Isonet and Rak dispensers were removed weekly from the incubators (Fratelli Galli, Biolog-Lux model, Milan, Italy), individually weighted (W, in g) with a precision balance (Sartorius BP 221S, Goettingen, Germany; weighing capacity measuring range: 220 g, precision: 0.1 mg, resolution: 0.1 mg, linearity: 0.2 mg), and finally relocated in the incubators. Each measurement lasted about 5 s, so the temperature variation between the incubator and the measurement room could be considered uninfluential. The measurements were repeated until dispensers weighed close to 1 ± 0.01 g for Isonet and 4.5 ± 0.05 g for Rak.

The dispensers kept at 30 °C were incubated in an oven (G-Cell 035, Fratelli Galli, Milan, Italy) at 60 °C at the end of the experiment to obtain the dispenser tare (i.e., the weight of the plastic material only). The dispensers were weighed daily and individually until there were no more changes in the weight between two consecutive measurements. The experiment was repeated once. The net weight (*NW*) of pheromones for each type of dispenser at any weighing time and any temperature was calculated as follows:(1)$${NW}_{t}=W_{t}-\bar{\tau }$$
where *W_t_* is the dispenser weight (g) at each weighing time and $\bar{\tau }$ is the average dispenser tare (g).

Any *NW_t_* was rescaled by its initial weight (*NW*_0_, i.e., the weight at the start of the experiment) on a 0 to 1 scale to determine the effect of temperature on the pheromone release. For instance, one of the five samples of Isonet incubated at 5 °C had *NW_t_*_0_ = 0.525 g at the experiment start (16th May) and = 0.523 g seven days after (*NW_t_*_7_, 23rd May). Then, the rescaled *NW* was obtained after seven days as 0.523/0.525 = 0.996.

Different equations were fitted to the dynamics of the rescaled *NW* to mathematically describe the effect of temperature on the pheromone release (PR) over time, which is hereinafter referred as the pheromone release rate (PRR). The best equations were selected based on the Akaike information criterion (AIC), which is an estimator of the relative quality of statistical models for a given set of data [26]. The PRR was calculated for each sample as the slope of the linear regression equation obtained by fitting the rescaled *NW* values over time.

Equation parameters were estimated using the function *nls* of the “stats” package of R software (R 4.0.3, 2020). The parameterized equations were evaluated for goodness of fit based on the adjusted R^2^ (R_adj_^2^), the root mean square error (RMSE), the coefficient of residual mass (CRM), and the concordance correlation coefficient (CCC) [27,28]. The adjusted R^2^ was estimated by conducting a linear regression between the observed and model-predicted values. Linear regression was performed using the *lm* function of the R “stats” package (R, 2020). The RMSE was obtained using the *rmse* function of the R “modelr” package [29]. The CCC was obtained using the CCC function of the R “DescTools” package [30]. In brief, RMSE represents the average distance of real data from the fitted line, and CRM is a measure of the tendency of the equation to overestimate or underestimate the observed values (a negative CRM indicates a tendency of the model toward overestimation) [28]. CCC is the product of the Pearson correlation coefficient and coefficient Cb, which indicates the difference between the best-fitting and perfect agreement lines (CCC = 1 means perfect agreement) [31].

### 2.3. Pheromone Release Under Vineyard Conditions

The experiment was conducted in 2017 and 2018 in an experimental vineyard at the Università Cattolica del Sacro Cuore campus (Piacenza, PC, 45°2′15.4536″ N, 9°43′46.326″ E; 80 m above sea level [a.s.l.]) and in 2018 in two commercial vineyards of the Colli piacentini denomination, at Castell’Arquato (CA, 44°51′26.0352″ N, 9°51′17.9748″ E; 400 m above sea level [a.s.l.]), and Vicobarone (VB, 44°59′31.542″ N, 9°21′28.0656″ E; 375 m above sea level [a.s.l.]). The vines at PC were two years old in 2017; they were trained using a Guyot system with within- and between-row spacings of 0.9 and 1 m, respectively. At CA, the vineyard was 11 years old in 2018, and plants were trained using a Guyot system. The within- and between-row spacings were 1.0 and 2.3 m, respectively. At VB, plants were two years old in 2018 and trained using a Guyot system; the within- and between-row spacings were 1.2 and 3 m, respectively.

In each vineyard, hourly data of temperature (T, in °C), relative humidity (RH, in %), rainfall (R, in mm), and wind speed (WS, in m/s) were recorded using an automated weather station (iMeteos; Pessl Instruments GmbH, Weiz, Austria) located < 1 km from the experimental plots. The growth stages of vines were assessed weekly in the vineyards according to the scale of Lorenz et al. [32].

At the PC vineyard, five dispensers per type (Isonet and Rak) were hung in the vineyard on 23 May 2017 and on 25 May 2018. At VB and CA vineyards, 90 dispensers per type were hung on 25 May 2018. The dispensers were collected at 7-day intervals, transported to the laboratory in a fridge, and weighed as described before. At the PC vineyard, considering the proximity of the vineyard to the laboratory, the dispensers were relocated soon after weighing and collected again 1 week after. At the VB and CA vineyards, 5 dispensers per type were collected at each sampling date, and after the weight measure, they were discarded as plant protection products. The sampling period was between 125 and 151 days based on the year. The tare of each dispenser was determined at the end of the sampling period, as previously described.

The NW of dispensers at any sampling time in any vineyard and year was calculated and rescaled to estimate the PR (on a 0 to 1 scale), as a relative reduction in dispenser NW, as described before. The rescaled data were then fit through a linear regression with the following general equation:PR = α + β_1_X_1_ + β_2_X_2_ + … + β_n_X_n_ + ε(2)
where α and β_1_ to β_n_ are equation parameters, ε is the error, and X_1_ to X_n_ are the following independent variables: (i) day of the year (DOY), (ii) accumulated values of PRR (ΣPRR) from the experiment under controlled conditions, (iii) degree days accumulated when temperature > 0 °C (ΣDD, in °C), (iv) accumulated daily precipitation (ΣDP, in mm), and (v) accumulated daily vapor pressure deficit (ΣVPD, in kPa) with the VPD being calculated as follows [33]: VPD = 0.61121 exp((18.678 − T/234.5)(T/(257.14 + T)))(1 − RH/100). Hourly meteorological data were transformed to obtain the daily averages of temperature and relative humidity. In (ii) and (iii), we were interested in evaluating the dynamics of PR over a temperature-dependent physiological time [34]. In (iv) and (v), we searched for a possible effect of rain or evaporative atmospheric potential on PR [35].

In the first set of equations, the independent variables were first used individually in a linear (i.e., with the term X_n_ only) and polynomial (i.e., with X_n_ and X_n_^2^). In the second set, the independent variables and their squared values were used altogether in a stepwise regression, which is a step-by-step method for adding and removing terms from a multilinear model based on their statistical significance in a regression. At each step, the independent variable not in the equation with the smallest probability of F is entered if that probability is <0.05; variables already in the regression equation are removed if their probability of F becomes >0.10. The method terminates when no more variables are eligible for inclusion or removal.

In a third set of equations, four dichotomous variables (with values 0 or 1) accounting for single vineyards (PC17, PC18, CA18, and VC18) were also included in a stepwise regression. Such inclusion was made to verify whether there are vineyard-specific conditions that are not related to the time (DOY) and weather-related variables (ΣPRR, ΣDD, ΣDP, and ΣVPD).

Equation parameters were estimated using the function *nls* of the “stats” package of R software (R, 2020). The parameterized equations were evaluated based on the standard error of estimates (SE_est_), R_adj_^2^, AIC, RMSE, CRM, and CCC. Only relevant equations were reported in the following sections.

## 3. Results

### 3.1. Effect of Temperature on Pheromone Release

For both dispensers, the highest weight loss was obtained at 30 °C with a difference between the average initial weight (W_i_) and the average final weight (W_f_) of 0.480 g and 0.337 g for Isonet and Rak, respectively (Table 1). The lowest weight loss was observed at 5 °C with values of 0.038 g and 0.014 g for Isonet and Rak, respectively.

The net weights of both dispensers decreased over the sampling date for about 120 days; no further weight changes were observed afterward. However, the slope at which dispensers lost weight differed at the selected temperatures (Figure 1A,B). Therefore, the slopes of the weight reductions differed (Table 1).

The dynamic of PRR (i.e., the pheromone release rate over time) for each dispenser was finally calculated using the following exponential equation:(3)$$PRR=\alpha e^{\beta T}$$
where *α* and *β* are equation parameters representing the lower limit of PRR and the intrinsic rate of increase in response to temperature (T, in °C) (Figure 2A,B).

Equation (3) provided a good data fit (Table 2).

### 3.2. Pheromone Release Under Vineyard Conditions

Weather conditions during the use of pheromone dispensers in vineyards were quite variable. At PC in 2017, WS remained almost constant with some peaks above 4 m/s (Figure 3A). The total amount of rain during the experimental period was 314 mm with peaks > 30 mm on June 14 (32.2 mm), July 11 (31.8 mm), July 24 (49.4 mm), July 29 (36.8 mm), and September 10 (89.6 mm). The daily temperature ranged from 14.4 to 30.8 °C with an overall average of 24.2 °C. The average VPD was 1.3 kPa and ranged from 0.3 to 2.3 kPa (Figure 3B).

At PC in 2018, WS was similar to that observed in 2017 with fewer peaks above 4 m/s (Figure 4A). However, the weather was a little bit dryer and cooler than 2017. Indeed, there was 274.7 mm of rain during the experimental period, and the daily temperature ranged from 12.2 to 30.4 °C with an average of 23.1 °C. The VPD ranged from 0.2 to 2.2 kPa with an overall average of 1 kPa (Figure 4B).

At CA in 2018, WS was constantly close to 0 with a maximum of 2.5 m/s on 16th September (Figure 5A). A total of 208.7 mm of rain fell during the experimental period. The daily temperature ranged from 13.2 to 31 °C, and the average was 24.4 °C. The average daily VPD was 1.2 kPa, ranging from 0.3 to 2.3 kPa (Figure 5B).

At VB in 2018, three days above 5 m/s were registered: September 24 (5.1 m/s), September 25 (5.7 m/s) and September 29 (5.0 m/s) (Figure 6A). In total, 175.4 mm of rain fell during the experimental period. The daily temperature ranged from 12.9 °C to 29.6 °C, and the average was 23.5 °C. The average daily VPD was 1.2 kPa, ranging from 0.2 to 2.4 kPa (Figure 6B).

The dynamics over time of PR by Isonet and Rak dispensers in the four vineyards are shown in panels C of Figure 3, Figure 4, Figure 5 and Figure 6. In general, NW loss has a linear trend with no peaks of decline.

For Isonet, the linear regression with ΣPRR as an independent variable provided the best fit of PR (Table 3); the use of ΣDD also provided a good fit, and both were more accurate than DOY (for instance, AIC decreased by 25 and 21, respectively), indicating that temperature had a relevant role in PR.

Regression with ΣDP and ΣVPD provided a worse fit than temperature-related variables (Table 3), indicating that rain and VPD had poor explanatory power when used singly. The use of square terms through polynomial regression for individual independent variables never showed a significant contribution, meaning that the decline in pheromone release over the selected independent variables was linear (Figure 7A).

Two variables, ΣPRR and ΣVPD, were linearly combined when all independent variables were used in stepwise regression, providing a relevant increase in goodness-of-fit (Table 3) and precision of estimates. The SE_est_ was 0.046 for the regression with ΣPRR only and 0.032 for the regression with ΣPRR and ΣVPD (Table 3). Therefore, the evaporative potential of the atmosphere (namely, the difference between the amount of moisture in the air and how much moisture the air can hold when saturated) played a relevant role in combination with temperature. Dichotomic variables related to single vineyards were never significantly included in the regression equation, indicating no considerable differences among vineyards.

For Rak, the best fit among the regressions with single independent variables was obtained with ΣVPD (Table 4) even though the differences among variables were less evident than for Isonet (cfr. Table 3).

Unlike Isonet, the quadratic term in polynomial regressions was always notable, meaning that the reduction in pheromone release was not linear (Figure 7B, referring to VPD). As for Isonet, the stepwise regression selected two linearly combined variables, ΣPRR^2^ and ΣVPD, which provided some increase in the goodness-of-fit, which remained lower than for Isonet (Table 4). However, the fit to real data significantly increased (e.g., the AIC went from −327 to −480), and the estimates were more accurate (e.g., the SE_est_ went from 0.111 to 0.039) when the dichotomic variable for vineyards was included in the analysis.

The comparison between predicted values based on the above regression equations and observed values of pheromone release is shown in Figure 8. The differences between the distribution of predicted and observed values are shown in Figure 8B,D, for Isonet and Rak, respectively. For both dispensers, the interquartile range is between −0.05 and 0.05, with the average close to 0, stating the good prediction of the chosen equations.

## 4. Discussion

MD has traditionally been used as an environmentally friendly method for controlling *L. botrana* and *E. ambiguella* using different dispensers for specific pheromones. These include sprayable formulations [36], female-equivalent dispensers [37], automatic aerosol devices [38], and hand-applied passive dispensers [39,40]. Among these dispensers, we selected the passive dispensers Isonet and Rak because they have been widely used to control EGVM and EGBM [19,41,42,43]. These dispensers are considerably different in shape, and it is known that dispenser shape and material influence the pheromone release [44].

Isonet consists of two parallel, thin ropes of polymeric material, one filled with pheromone welded at the ends and open in the center to allow application on the plant. Isonet technology allows pheromone release to be constant and regular during the season. The dispenser wall, which is characterized by many small pores, remains impregnated with the active substance by capillarity throughout the season even when its content is decreasing; this ensures a constant flow of pheromones from the diffuser to the environment [25]. The Rak dispenser is made of porous plastic with two ampoules containing the active substance. In this type of dispenser, the liquid pheromone permeates outside by exploiting the characteristics of the plastic material in which it is contained. Once on the external wall, the pheromone volatilizes and spreads into the environment [44,45].

The different shapes of the two dispensers may have influenced the PR patterns observed [46]. Isonet has an approximate surface area of A = 17.2 cm^2^ and a volume of V = 1.69 cm^3^, which results in a surface area-to-volume ratio of 10.15 (1/cm); Rak has A = 20.1 cm^2^, V = 6.97 cm^3^, with a surface area-to-volume ratio of 2.88 (1/cm) [47]. This ratio was 3.5 times higher in Isonet than in Rak, affecting pheromone release dynamics. The porosity (e.g., type, number per surface unit, and size of pores) of the dispenser materials may also influence the release, but details on these characteristics are not available. It may also be considered that Rak contains a substance (*N*-dodecenyl acetate) that delays pheromone release (which is absent in Isonet), which could, to some extent, influence the pheromone release dynamic over time.

For both dispensers, temperature and VPD explained the PR in vineyards characterized by diverse weather conditions better than time; this is consistent with the mechanism of pheromone evaporation from the dispenser surface and its consequent rewetting by capillarity. Indeed, VPD essentially represents how easily water evaporates; a low VPD means that water evaporates slowly, while increasing VPD indicates increased evaporation. Low temperatures yielded a low VPD at a fixed RH, while high temperatures resulted in a high VPD [48]. Therefore, pheromone evaporation from the dispenser surface may be higher at high VPD than at low VPD. Additionally, as the temperature of the pheromone increases, its surface tension diminishes, and this facilitates attractive forces between pheromone molecules and dispenser walls to pull the internal liquid into capillary pores and wet pheromone molecules out [49]. In our preliminary experiment, the effect of temperature on PRR was not linear, and any temperature increase at high temperatures accelerated the release rate more than proportionally. Zhu et al. [24] obtained similar results with dispensers different from the ones we used.

Weather conditions, particularly temperature, have been previously observed to potentially influence pheromone release [50,51,52,53]. For instance, Moschos et al. [54] observed that pheromone release by the Rak 2 Plus dispenser in a 3-year period (1996–1998) was high in July and minimal in April and September–October due to hot temperatures in the summer months. Arn et al. [50] found that the pheromone release rate by the Rak 1 + 2 ampoule increased by a factor of 10 from 15 to 35 °C and observed that this temperature-dependent release does not match the male activity of moths, which was high between dusk and the end of the scotophase when temperatures were low [2,50]. Van der Kraan and Ebbers [55] noticed a 2–2.5-fold increase in the pheromone emission by polythene tubes with a rise in temperature between 15 °C and 25 °C. Our results are in agreement with previous studies. However, the exponential function we used for studying the temperature-dependent PRR under field conditions better fits real data than degree days, confirming the non-linear effect of temperature.

The effect of wind and RH has been considered of no or minor importance in previous studies [23,56,57] even though there was evidence of the effect of certain wind speeds on the increase in pheromone release [23,46,55,58] and of high humidity on the release reduction [24,59,60] under environment-controlled conditions. Indeed, increased wind speeds may facilitate the volatilization of pheromones from dispenser surfaces, thus increasing the mobility of the inner molecules [23,24,44,46,61]. Additionally, high environmental humidity may have the opposite effect. The considerable impact of VPD on PR in our study under vineyard conditions would indirectly support the role of wind and humidity, as RH is directly related to VPD, and wind is related to RH levels [62]. In another field study, Gavara et al. [63] found that daily pheromone release rates by an Isonet dispenser were predicted (with R^2^ = 0.378) by a multiple regression that included minimum and maximum daily temperatures and average WS. Precipitation had no notable effect in our study nor the study of Gavara et al. [63].

In contrast to Isonet, Rak showed a non-linear release trend of the pheromone emission in all tested field conditions; further, the stepwise regression pointed out the importance of the dichotomous variable of the vineyard to increase goodness of fit. This effect may be due to the pheromone-absorbing capacity of the canopy, as shown in several publications [64]. The waxy surface of the leaves seems to be the site responsible for the uptake and release of pheromones by plants, considering also that the ground cover might affect in minimal part the absorption of the pheromone [21]. Gavara et al. [25] suggested that the plant canopy is unnecessary to achieve MD with passive dispensers. Still, it may keep the pheromone in the field area, leading to a more uniform distribution and an increase in the number of pheromone sources along the crop, consequently improving the competitive mechanisms of disruption. The MD competitive mechanism, i.e., false trails formation, is achieved when artificial sources emitting pheromones overcome plumes produced by adult females. On the other hand, a non-competitive mechanism, i.e., the desensitization of males, typically takes place when moths are exposed to huge concentrations of pheromones, which is hardly obtained in vineyards treated with MD dispensers [65,66]. Studies have shown that the competitive mechanism is more exerted in MD control for EGVM. On the contrary, EGBM females produce a very attractive blend of pheromones, critically influencing the competitive mechanism in MD [19].

Our results were obtained by estimating PR by measuring the weight loss of dispensers over time, as in previous studies (e.g., [22,67,68]). It could then be questioned if this method reflects the emission of pheromones into the vineyard air, in case the dispenser weight loss was caused by the loss of other volatile components eventually present in the dispenser, by the decomposition of some dispenser materials, or because precipitation and moisture influence the absorption and desorption of water by dispensers [22]. An alternative method is extracting residual pheromone from the dispenser using gas chromatography [11]. In a comparative study under field conditions, however, Il’ichev et al. [36] found that the weight loss method and gas chromatography provided comparable results. The weight loss method is non-destructive, does not require expensive equipment, and is cheaper. Additionally, it can be easily used in routine evaluations within MD programs, increasing the exploitation of our results. Our results indicate that weather conditions influencing pheromone release (namely, air temperature and VPD) may be common for different dispenser types [23]. However, our equations were dispenser-dependent, so equations relating these influencing variables to the pheromone release pattern must be defined for each dispenser type.

## 5. Conclusions

The equations developed in this work can potentially be used in the management of pheromone dispensers for MD. Effective MD requires dispensers application before the first flight of the moths [69] and the coverage of all flights until the harvest, which are especially critical with late season varieties [63]. These equations, eventually linked to mathematical models predicting the phenology of EGVM [70] and EGBM [71] populations, may support the farmers in advance about the right moment for the dispenser’s application and the pheromone release available to disrupt during flights. Indeed, these equations can provide information on the PR during the season, as influenced by weather conditions. Since the dispensers need to ensure the release of a sufficient pheromone dose to disrupt the mating of the target pests across different pest generations, these equations could serve as an input for an atmospheric pheromone concentration model to predict concentrations based on meteorological conditions [72]. However, the equations should be further validated against independent data collected under diverse environmental conditions before use. These validation efforts should also consider the efficacy of MD through pheromone traps for monitoring the adult male population and sampling eggs, larvae, and damaged barriers.

## Figures and Tables

**Figure 1 insects-15-00962-f001:**
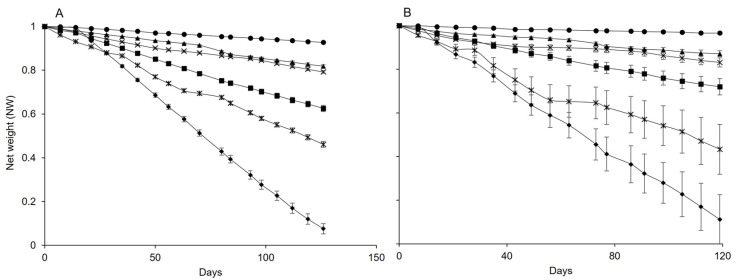
Changes over time in the net weight of pheromone dispensers Isonet (**A**) and Rak (**B**) at different temperatures: 5 °C (circles), 10 °C (full triangles), 15 °C (cross), 20 °C (square), 25 °C (stars), and 30 °C (diamond); whiskers show the standard error.

**Figure 2 insects-15-00962-f002:**
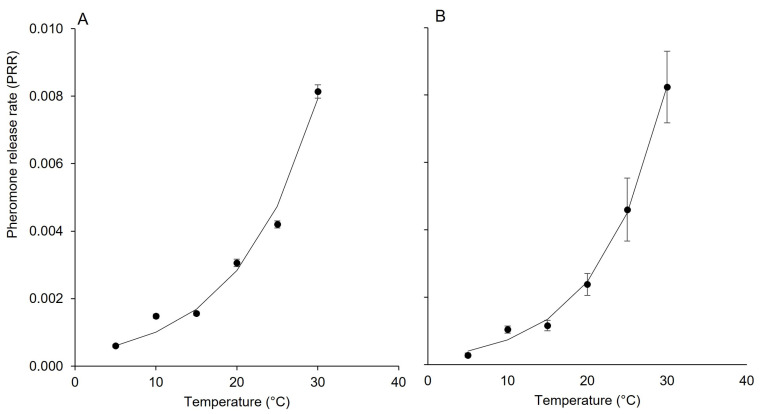
Changes in the pheromone release rate (PRR) as affected by temperature for pheromone dispensers Isonet (**A**) and Rak (**B**); whiskers show the standard error.

**Figure 3 insects-15-00962-f003:**
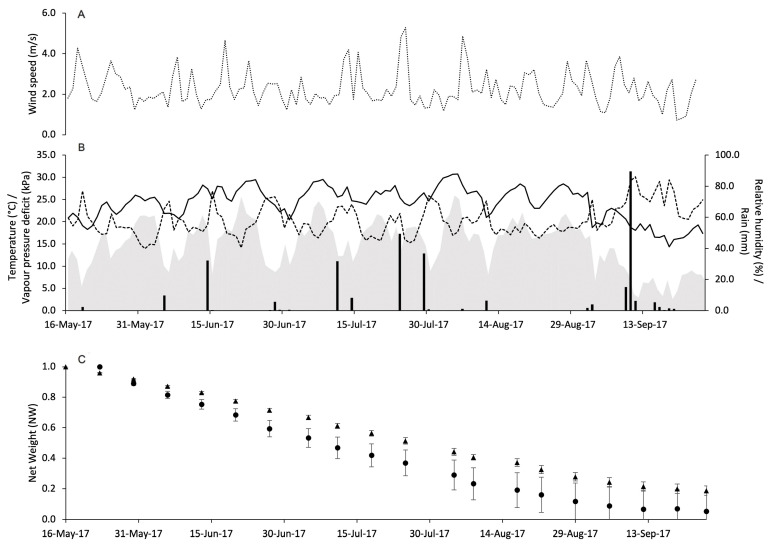
Weather conditions registered in the vineyard of Piacenza in 2017: wind speed (m/s) (**A**), air temperature (°C, full line), relative humidity (%, dotted line), rainfall (mm, black bars), and VPD (kPa, gray area) (**B**). In (**C**), changes in the net weight of pheromone dispensers Isonet (circles) and Rak (triangles) exposed into the vineyard; whiskers show the standard error.

**Figure 4 insects-15-00962-f004:**
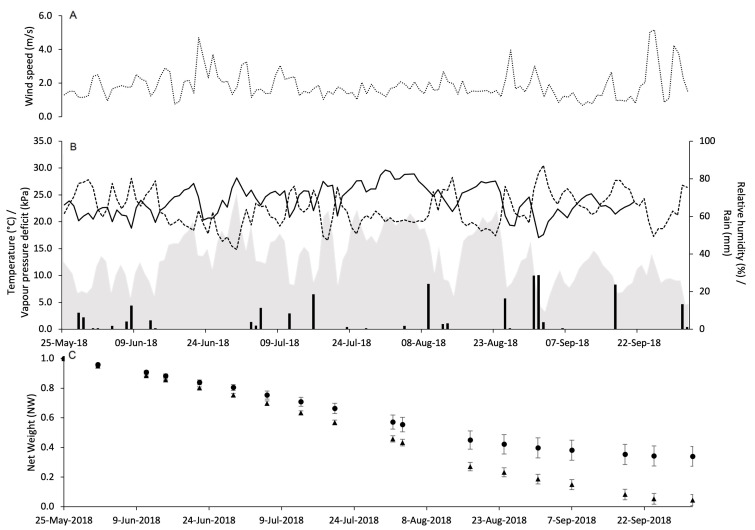
Weather conditions registered in the vineyard of Piacenza in 2018: wind speed (m/s) (**A**), air temperature (°C, full line), relative humidity (%, dotted line), rainfall (mm, black bars), and VPD (kPa, gray area) (**B**). In (**C**), changes in the net weight of pheromone dispensers Isonet (circles) and Rak (triangles) exposed into the vineyard; whiskers show the standard error.

**Figure 5 insects-15-00962-f005:**
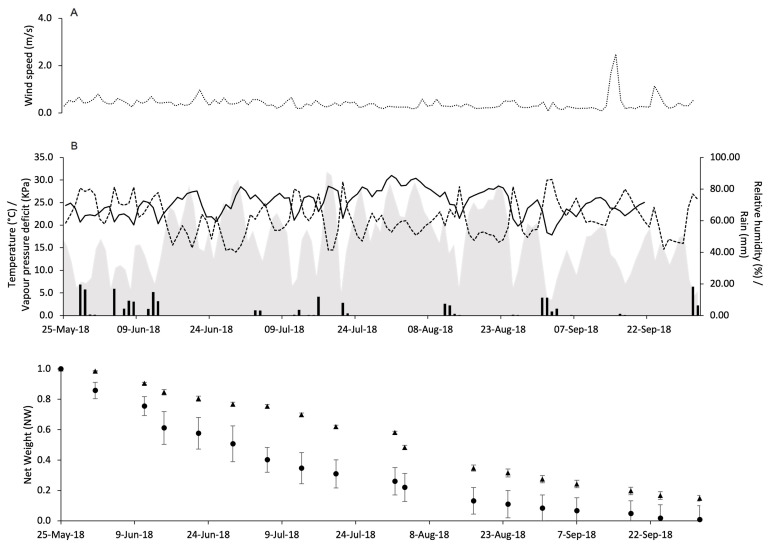
Weather conditions registered in the vineyard of Castell’Arquato in 2018: wind speed (m/s) (**A**), air temperature (°C, full line), relative humidity (%, dotted line), rainfall (mm, black bars), and VPD (kPa, gray area) (**B**). In (**C**), changes in the net weight of pheromone dispensers Isonet (circles) and Rak (triangles) exposed into the vineyard; whiskers show the standard error.

**Figure 6 insects-15-00962-f006:**
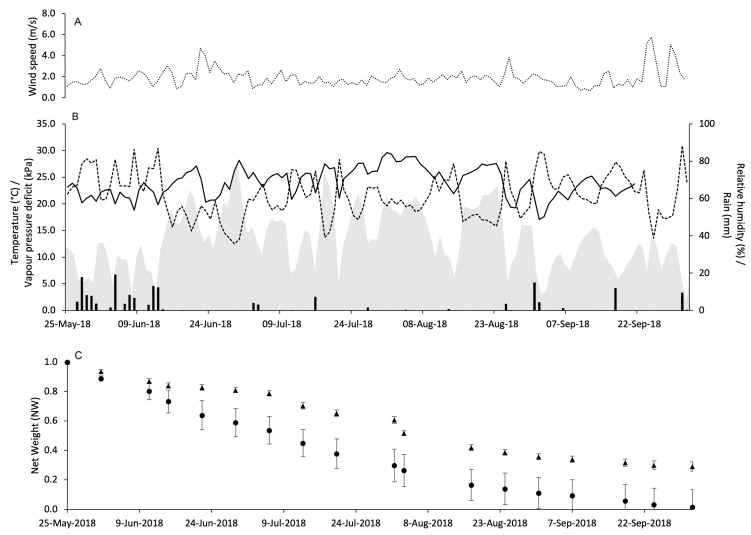
Weather conditions registered in the vineyard of Vicobarone in 2018: wind speed (m/s) (**A**), air temperature (°C, full line), relative humidity (%, dotted line), rainfall (mm, black bars), and VPD (kPa, gray area) (**B**). In (**C**), changes in the net weight of pheromone dispensers Isonet (circles) and Rak (triangles) exposed into the vineyard; whiskers show the standard error.

**Figure 7 insects-15-00962-f007:**
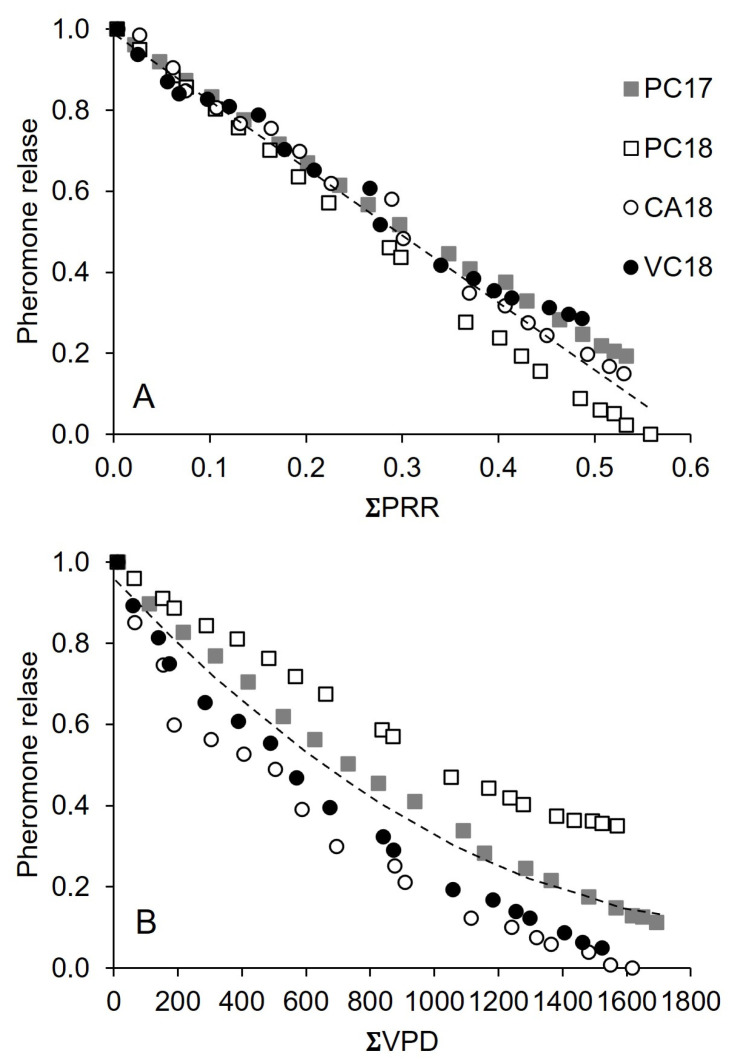
Pheromone release (as relative reduction in net dispenser weight) by Isonet (**A**) and Rak (**B**) dispenser as a function of accumulated, temperature-dependent pheromone release rate (ΣPRR) and accumulated vapor pressure (ΣVPD, kPa), respectively, in four vineyards: Piacenza in 2017 (PC17) and 2018 (PC18), Castell’Arquato (CA18) and Vicobarone (VB18) in 2018. The dotted lines show the regression equations in Table 3 (line 2) and Table 4 (line 5) for Isonet and Rak, respectively.

**Figure 8 insects-15-00962-f008:**
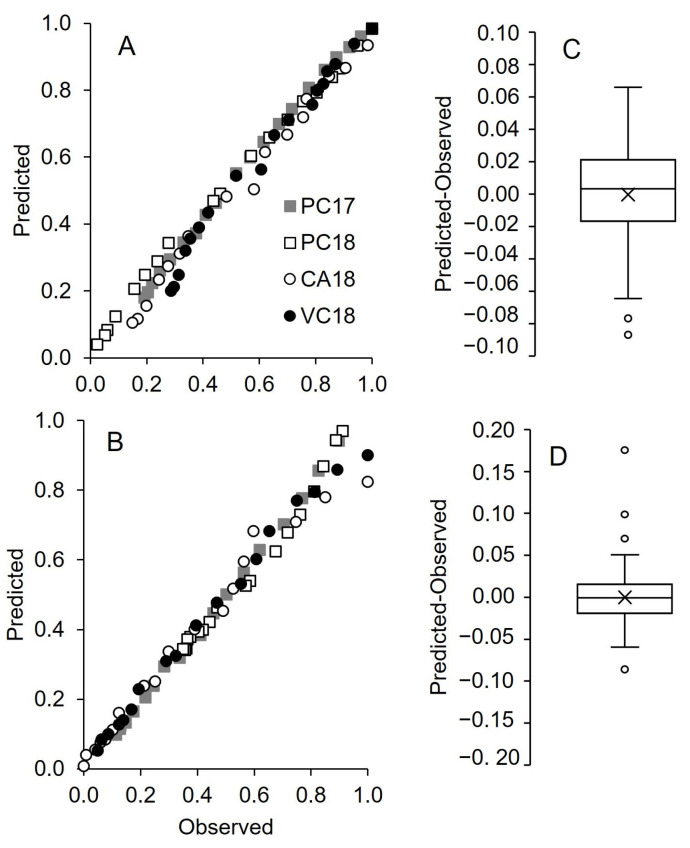
Comparison between predicted and observed pheromone release (as relative reduction in net dispenser weight) by Isonet (**A**,**C**) and Rak (**B**,**D**) dispensers in four vineyards: Piacenza in 2017 (PC17) and 2018 (PC18), Castell’Arquato (CA18) and Vicobarone (VB18) in 2018. Predicted values were obtained through the regression equations in Table 3 (line 6) and Table 4 (line 7) for Isonet and Rak, respectively. Box plots show the distribution of differences between predicted and observed; the line crossing the boxes represents the median, and × indicates the average; whiskers extend to the maximum and minimum.

**Table 1 insects-15-00962-t001:** Average values of initial and final weights of 5 pheromone dispensers per type (Isonet and Rak) kept at different temperatures (T) for ~120 days.

T (°C)	Isonet					
	*W_i_* ^a^	*W_f_* ^b^	*τ* ^c^	*NW_f_* ^d^	Slope ^e^	R_adj_^2, f^
5	1.405	1.367	-	0.927	0.004	0.998
10	1.382	1.291	-	0.819	0.011	0.989
15	1.401	1.292	-	0.792	0.011	0.989
20	1.8	1.203	-	0.625	0.022	0.997
25	1.398	1.117	-	0.461	0.030	0.996
30	1.396	0.917	0.876	0.076	0.058	0.997
**T (°C)**	**Rak**					
	***W_i_* ^a^**	***W_f_* ^b^**	***τ* ^c^**	***NW_f_* ^d^**	**Slope ^e^**	**R_adj_^2, f^**
5	4.878	4.864	-	0.966	0.002	0.977
10	4.940	4.877	-	0.872	0.007	0.984
15	4.932	4.852	-	0.831	0.008	0.928
20	4.859	4.748	-	0.720	0.017	0.996
25	4.787	4.618	-	0.434	0.033	0.976
30	4.841	4.503	4.434	0.111	0.054	0.998

^a^ Average initial weight (g) of the dispenser; ^b^ Average final weight (g) of the dispenser; ^c^ Average tare weight (g) of dispenser; ^d^ Average final net weight of dispenser Equation (1); ^e^ Slope of linear regression obtained by fitting the average rescaled values over time; ^f^ R_adj_^2^ of linear regression obtained by fitting the average rescaled values over time.

**Table 2 insects-15-00962-t002:** Parameters of Equation (3) fitting the pheromone release rate of two dispensers concerning temperature and statistics for goodness-of-fit to real data.

Dispenser	Parameters ^a^		Statistics ^b^			
	*α*	*β*	R_adj_^2^	RMSE	CRM	CCC
Isonet	3.597 × 10^−4^	1.031 × 10^−1^	0.980	3.224 × 10^−4^	1.258 × 10^−2^	0.992
Rak	2.172 × 10^−4^	1.213 × 10^−2^	0.995	1.620 × 10^−4^	4.924 × 10^−4^	0.998

^a^ Estimates of parameters of Equation (3); ^b^ adjusted R^2^, root mean square error (RMSE), coefficient of residual mass (CRM), concordance correlation coefficient (CCC).

**Table 3 insects-15-00962-t003:** Parameters of the regression equation fitting the dynamics of pheromone release from Isonet dispensers based on different independent variables used singly (lines 1 to 5) or combined in a stepwise regression (line 6), criteria for the selection of the best regression equation, and indexes of goodness-of-fit to real data.

X ^a^	Parameters		Selection Criteria ^b^			Goodness-of-Fit ^c^		
	*α*	*β*	SE_est_	R_adj_^2^	AIC	RMSE	CRM	CCC
1. DOY	1.986	−0.007	0.055	0.965	−433	0.061	0.026	0.979
2. ΣPRR	0.993	−1.662	0.046	0.975	−458	0.047	0.003	0.987
3. ΣDD	1.006	−0.00028	0.048	0.973	−454	0.050	0.004	0.985
4. ΣDP	0.955	−0.00333	0.145	0.758	−288	0.142	−0.002	0.865
5. ΣVPD	0.984	−0.00053	0.075	0.935	−387	0.074	0.002	0.967
6. ΣPRR and ΣVPD	0.089	−3.441 0.00058	0.032	0.988	−522	0.031	0.001	0.994

^a^ Independent variables: DOY, day of the year; (ii) ΣPRR, accumulated pheromone release rate; ΣDD, degree days accumulated when temperature > 0 °C; ΣDP, sum of daily precipitation; ΣVPD, accumulated daily vapor pressure deficit; ^b^ SE_est_, standard error of estimates; R_adj_^2^, adjusted R^2^; AIC, Akaike information criterion; ^c^ RMSE, root mean square error; CRM, coefficient of residual mass; CCC, concordance correlation coefficient.

**Table 4 insects-15-00962-t004:** Parameters of the regression equation fitting the dynamics of pheromone release from Rak dispensers based on different independent variables used singly (lines 1 to 5) or combined in a stepwise regression (lines 6 and 7), criteria for the selection of the best regression equation, and indexes of goodness-of-fit to real data.

X ^a^	Parameters ^b^			Selection Criteria ^c^			Goodness-of-Fit ^d^		
	*α*	*β* _1_	*β* _2_	SE_est_	R_adj_^2^	AIC	RMSE	CRM	CCC
1. DOY	3.885	−0.027	0.049	0.121	0.832	−314	0.114	−0.008	0.918
2. ΣPRR	0.982	−3.126	3.034	0.123	0.827	−311	0.115	−0.002	0.915
3. ΣDD	1.014	−0.005	0.00008	0.121	0.832	−314	0.114	−0.002	0.917
4. ΣDP	1.026	−0.0068	0.013	0.140	0.776	−292	0.136	0.002	0.879
5. ΣVPD	0.963	−0.0008	0.00021	0.118	0.840	−317	0.110	−0.001	0.923
6. ΣPRR^2^ ΣVPD	0.974	2.456 −0.0009	−	0.111	0.859	−327	0.102	−0.001	0.935
7. ΣPRR^2^ ΣVPD PC17 CA18 VC18	1.095	2.171 −0.0009 −0.059 −0.260 −0.185	−	0.039	0.982	−480	<0.001	−0.001	0.999

^a^ Independent variables: DOY, day of the year; (ii) ΣPRR, accumulated pheromone release rate; ΣDD, degree days accumulated when temperature > 0 °C; ΣDP, sum of daily precipitation; ΣVPD, accumulated daily vapor pressure deficit; PC17, CA18 and VC18, dichotomic variables accounting for the correspondent vineyards; ^b^
*β*_2_ refers to the quadratic term for regression with single independent variables and to ΣVPD for the stepwise regression; ^c^ SE_est_, standard error of estimates; R_adj_^2^, adjusted R^2^; AIC, Akaike information criterion; ^d^ RMSE, root mean square error; CRM, coefficient of residual mass; CCC, concordance correlation coefficient.

## Data Availability

The raw data supporting the conclusions of this article will be made available by the authors without undue reservation.

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
