# Peer review of "The Dynamics of Pheromone Release in Two Passive Dispensers Commonly Used for Mating Disruption in the Control of Lobesia botrana and Eupoecilia ambiguella in Vineyards"

_insects, 2024, doi:10.3390/insects15120962_

Round 1
Reviewer 1 Report
Comments and Suggestions for Authors
General remark:
This is an interesting and valuable study on a topic (impact of weather conditions) that is very important for the application of mating disruption, but so far little studied and known. However, I have some concerns about the research design and general conclusions in this study.
As PRR (= 𝛼𝑒𝛽𝑇) and day degrees (DD) are calculated based on only temperature (and hence only dependent on temperature), it seems logical that the pheromone release is dependent on DD as well as PRR. This was also the main outcome of the regression analysis with single variables (Table 3 and 4). These variables are not independent from each other, and I was wondering if regression analysis with variables that are not independent from each other makes sense? Can the authors argue why this analysis still makes sense (or not)?
Furthermore, also VPD is partly dependent on temperature (VPD = 0.61121 exp((18.678 − T/234.5)(T/(257.14 + T)))(1 − RH/100)). In line with this it was also found to be a significant ‘independent’ variable in the analysis for RAK (Table 4) but in the analysis for Isonet it provided a worse fit. This would mean that in the case of Isonet the relative humidity (RH) worsened the effect of T? I was wondering why RH was not included as independent variable in the regression analysis? Can the authors argue on this? If there is no good reason, I suggest that the analysis should be repeated with RH (or accumulated RH?) as independent variable. I would expect that RH is (at least partly) dependent on the precipitation, so it would be interesting to figure this out, in order to also confirm the (absence of the) role/impact of precipitation. After all, in the described analysis no notable effect of precipitation was found, but this seems contradictory to the fact that a notable effect for VPD (depending partly on RH) was found. Or is it really only the temperature that is the decisive variable in this?
Please clarify, or redo the analysis with RH as independent variable.
Specific remarks:
Abstract:
Line 31: One dispenser type showed a non-linear release trend of the pheromone emission in field conditions.
Non-linear release trend in function of what? This is unclear and should be specified to make the abstract understandable. Based on the outcomes of this study it should be “a non-linear release trend of the pheromone emission in function of time (day of the year), accumulated (temperature dependent) pheromone release rate and degree days (temperature > 0°C).“
Line 33: dichotomous variable of the vineyard.
Dichotomous variables are categorical (nominal) variables which have only two categories or levels. However, in this study 4 different vineyards were investigated. So there are 4 categories or levels in the variable of the vineyard, and hence this is not a dichotomous variable. Or do I misunderstand and are there 4 dichotomous variables (for each vineyard one variable PC17, PC18, CA18, and VC18) with two levels (yes/no)? Please clarify and correct if necessary (also in rest of manuscript line 222, 362, 375, 399…)
1. Introduction:
OK, no questions, corrections or suggestions for improvements.
2. Materials and Methods
2.2 Effect of temperature on pheromone release
Line 130. Dispensers were placed in incubators at six constant temperature regimes
Which type of incubator was used? Was the relative humidity controlled (constant) in this incubators?
2.3 Pheromone release under vineyard conditions
Line 192: At VB and CA vineyards, 90 dispensers were hung. 90 dispensers per type or 45 dispensers per type? Or?
Line 193: The dispensers were collected at 7-days intervals
I assume all 10 (2x5) dispensers were collected each sampling time. Correct?
Line 196-197: dispensers were not relocated, and new dispensers were collected after seven days.
This is not clear. If I understand well not all 90 dispensers were sampled each time. How many? 2x5 (like in the PC vineyard) I assume. Correct?
Line 208: accumulated daily vapor pressure deficit (SVPD, in kPa) with VPD calculated as follows: VPD = 0.61121 exp((18.678 − T/234.5)(T/(257.14 + T)))(1 − RH/100)
This is not clear. You have hourly data from the weather stations. How was the daily VPD calculated? I assume for every hour VPD was calculated with the formula above (using the hourly measured T and hourly measured RH), and divided by 24 to have the hourly VPD, and subsequently, all 24 (hourly) VPD values of 1 day were summed to become the daily value. Correct? Or was it by first calculating the mean daily temperature and mean daily RH, and subsequently calculating the daily VPD based on these mean T and mean RH?
Line 211: In iii) and iv), we searched for a possible effect of rain or evaporative atmospheric potential
I am confused. I think this should be “in iv) and v), we searched for…”? Correct?
3. Results
Line 240 (Table 1). Are the values in the table the mean of:
- five dispensers that were used per type and temperature (Line 131)?
or
- 10 dispensers, as the experiment was repeated once (Line 142)?
Line 290: the weather was a little bit dry and cool
I think this should be: the weather was a little bit dryer and cooler than in 2017
Line 332: Table 3
Why is the first two rows in the table (1. DOY) identically repeated in the second row?
It is not clear for me how the variable SPRR was calculated.
In the materials and methods section (line 205) it is mentioned:
“accumulated values of PRR (SPRR) from the experiment under controlled conditions”
So I assume the exponential equation PRR = 𝛼𝑒𝛽𝑇 was used for this.
I assume that SPRR is an accumulation of daily calculated pheromone release rates? Correct?
But how exactly are the daily PRR’s calculated? Was it done by the exponential equation PRR = 𝛼𝑒𝛽𝑇 with daily mean temperatures? Or by first calculating the PRR for every hour (with the hourly temperatures)/24 and sum all 24 values to have the daily value? (and then accumulate the daily values to have the SPRR). Please clarify.
Line 343-344:
In general the number of heavy rain events was low. What would be the outcome if you had a very wet and rainy season as we have in times of climate change? do you expect the effect of rain and VP to be minor then?
Did you check the effect of single events? for example a rainy week? Or can the effect of rain completely be included in the VPD as rain affects RH?
Line 345: The use of polynomial regressions for individual independent variables never showed a significant contribution to the square terms,
I am confused. I think this should this rephrased as “contribution of the square terms” (?)
Line 346
meaning that the decline in pheromone release over time was linear.
This is a linear relationship between pheromone release and the relevant variables, which is not ‘time’, but DOY (day of the year) and SPRR (accumulated pheromone release rate). So in the sentence above ‘over time’ should be corrected/rephrased.
Line 366: “obtained SVPD” should be “obtained with SVPD”
Line 383: (Figure 8, referring to VPD)
I think this should be Figure 7B, referring to VPD
Consequently, in that case there is no reference to Figure 8 in the Results section (nor in the rest of the manuscript). Please add a (small) paragraph with the description of the outcomes of the comparison between predicted and observed pheromone release levels (Figure 8).
4. Discussion
Line 437
Additionally, as the temperature of the pheromone increases, its surface tension diminishes, and this facilitates attractive forces between pheromone molecules and other surfaces in the vicinity to pull water into capillary pores of the dispenser wall and wet them out.
This is unclear for me. Are the authors referring to the fact that precipitation and moisture influence the absorption and desorption of water by dispensers? Or what do you mean by “pull water into capillary pores of the dispenser wall and wet them out”?
Line 480
Considering the mechanism of mating disruption of EGVM and EGBM. Is it competitive or non-competitive disruption? Depending on the mechanism what is the most important: the concentration of pheromone at the surface of the dispensers (point sources) (competitive) or the concentration of pheromone in the air of the vineyard (non-competitive: desensitization of males). Using the type of passive dispenser, competitive attraction seems to be the main mechanism acting in disrupted fields (Gavara et al., 2020). I think this is an important element, so can you discuss this and include it in the discussion section?
Line 485: “precipitation and moisture influence the absorption and desorption of water by dispensers.”
However, precipitation had no notable effect on pheromone release in this study. How do you reconcile this with the discussion about “pull water into capillary pores of the dispenser wall and wet them out”? (see line 437 above)
Author Response
General remark:
This is an interesting and valuable study on a topic (impact of weather conditions) that is very important for the application of mating disruption, but so far little studied and known. However, I have some concerns about the research design and general conclusions in this study.
As PRR (= ????) and day degrees (DD) are calculated based on only temperature (and hence only dependent on temperature), it seems logical that the pheromone release is dependent on DD as well as PRR. This was also the main outcome of the regression analysis with single variables (Table 3 and 4). These variables are not independent from each other, and I was wondering if regression analysis with variables that are not independent from each other makes sense? Can the authors argue why this analysis still makes sense (or not)?
We don’t fully understand the comment. DD are commonly used to verify if a phenomenon is influenced by temperature; then, we used DD to verify if the pheromone release was influenced by temperature under field conditions. Similarly, we used a temperature-dependent function expressing the relation between pheromone release rate and temperature to verify if this function can represent the dynamic of pheromone release under field conditions. We agree that “it seems logical that the pheromone release is dependent on DD as well as PRR”, but our aim was to express this in mathematical terms.
Furthermore, also VPD is partly dependent on temperature (VPD = 0.61121 exp((18.678 − T/234.5)(T/(257.14 + T)))(1 − RH/100)). In line with this it was also found to be a significant ‘independent’ variable in the analysis for RAK (Table 4) but in the analysis for Isonet it provided a worse fit. This would mean that in the case of Isonet the relative humidity (RH) worsened the effect of T? I was wondering why RH was not included as independent variable in the regression analysis? Can the authors argue on this? If there is no good reason, I suggest that the analysis should be repeated with RH (or accumulated RH?) as independent variable. I would expect that RH is (at least partly) dependent on the precipitation, so it would be interesting to figure this out, in order to also confirm the (absence of the) role/impact of precipitation. After all, in the described analysis no notable effect of precipitation was found, but this seems contradictory to the fact that a notable effect for VPD (depending partly on RH) was found. Or is it really only the temperature that is the decisive variable in this?
Please clarify, or redo the analysis with RH as independent variable.
We did a preliminary analysis considering RH as sole influencing factors, but they did not provide a reliable fit. In the case of RAK the fit was worst but “not bad”, and we cannot say that RH worsened the effect of temperature in this case. In the discussion, we address the possible role of both rain and RH (lines 488-500).
Specific remarks:
Abstract:
Line 31: One dispenser type showed a non-linear release trend of the pheromone emission in field conditions.
Non-linear release trend in function of what? This is unclear and should be specified to make the abstract understandable. Based on the outcomes of this study it should be “a non-linear release trend of the pheromone emission in function of time (day of the year), accumulated (temperature dependent) pheromone release rate and degree days (temperature > 0°C).
Addressed (line 33: One dispenser type showed a non-linear release trend of the pheromone emission in field conditions respect to the considered variables).
Line 33: dichotomous variable of the vineyard.
Dichotomous variables are categorical (nominal) variables which have only two categories or levels. However, in this study 4 different vineyards were investigated. So there are 4 categories or levels in the variable of the vineyard, and hence this is not a dichotomous variable. Or do I misunderstand and are there 4 dichotomous variables (for each vineyard one variable PC17, PC18, CA18, and VC18) with two levels (yes/no)? Please clarify and correct if necessary (also in rest of manuscript line 222, 362, 375, 399…)
You are right: there are 4 dichotomous variables (for each vineyard one variable PC17, PC18, CA18, and VC18) with two levels (1/0). We added this in the text (line 231).
- Introduction:
OK, no questions, corrections or suggestions for improvements.
- Materials and Methods
2.2 Effect of temperature on pheromone release
Line 130. Dispensers were placed in incubators at six constant temperature regimes
Which type of incubator was used? Was the relative humidity controlled (constant) in this incubators?
Information included in the text (line 129).
2.3 Pheromone release under vineyard conditions
Line 192: At VB and CA vineyards, 90 dispensers were hung. 90 dispensers per type or 45 dispensers per type? Or?
The information was added in the text (lines 191-192).
Line 193: The dispensers were collected at 7-days intervals
I assume all 10 (2x5) dispensers were collected each sampling time. Correct?
Correct.
Line 196-197: dispensers were not relocated, and new dispensers were collected after seven days. This is not clear. If I understand well not all 90 dispensers were sampled each time. How many? 2x5 (like in the PC vineyard) I assume. Correct?
Correct. We modified the text (lines 200-202).
Line 208: accumulated daily vapor pressure deficit (SVPD, in kPa) with VPD calculated as follows: VPD = 0.61121 exp((18.678 − T/234.5)(T/(257.14 + T)))(1 − RH/100)
This is not clear. You have hourly data from the weather stations. How was the daily VPD calculated? I assume for every hour VPD was calculated with the formula above (using the hourly measured T and hourly measured RH), and divided by 24 to have the hourly VPD, and subsequently, all 24 (hourly) VPD values of 1 day were summed to become the daily value. Correct? Or was it by first calculating the mean daily temperature and mean daily RH, and subsequently calculating the daily VPD based on these mean T and mean RH?
It was obtained by first calculating the mean daily temperature and RH. An explanatory sentence was added (lines 217-218).
Line 211: In iii) and iv), we searched for a possible effect of rain or evaporative atmospheric potential
I am confused. I think this should be “in iv) and v), we searched for…”? Correct?
Correct. The reference was changed.
- Results
Line 240 (Table 1). Are the values in the table the mean of:
- five dispensers that were used per type and temperature (Line 131)?or
- 10 dispensers, as the experiment was repeated once (Line 142)?
The values are the mean of 5 dispensers used per type and temperature. It was specified (line 249).
Line 290: the weather was a little bit dry and cool
I think this should be: the weather was a little bit dryer and cooler than in 2017
The sentence was corrected (line 301).
Line 332: Table 3
Why is the first two rows in the table (1. DOY) identically repeated in the second row?
It is not clear for me how the variable SPRR was calculated.
In the materials and methods section (line 205) it is mentioned:
“accumulated values of PRR (SPRR) from the experiment under controlled conditions”
So I assume the exponential equation PRR = ???? was used for this.
I assume that SPRR is an accumulation of daily calculated pheromone release rates? Correct?
But how exactly are the daily PRR’s calculated? Was it done by the exponential equation PRR = ???? with daily mean temperatures? Or by first calculating the PRR for every hour (with the hourly temperatures)/24 and sum all 24 values to have the daily value? (and then accumulate the daily values to have the SPRR). Please clarify.
Doy repetition was a mistake.
SPRR was elaborated as an accumulation of daily pheromone release rate calculated daily as expressed in equation [3] and cumulated.
A clarification on weather daily data was added (line 217-218).
Line 343-344:
In general the number of heavy rain events was low. What would be the outcome if you had a very wet and rainy season as we have in times of climate change? do you expect the effect of rain and VP to be minor then?
Did you check the effect of single events? for example a rainy week? Or can the effect of rain completely be included in the VPD as rain affects RH?
This is an interesting question. We don’t have answer at this stage, and further research is needed, maybe by using rain simulators.
Line 345: The use of polynomial regressions for individual independent variables never showed a significant contribution to the square terms,
I am confused. I think this should this rephrased as “contribution of the square terms” (?)
The sentence was rephrased (lines 356-357).
Line 346
meaning that the decline in pheromone release over time was linear.
This is a linear relationship between pheromone release and the relevant variables, which is not ‘time’, but DOY (day of the year) and SPRR (accumulated pheromone release rate). So in the sentence above ‘over time’ should be corrected/rephrased.
The sentence was rephrased (lines 358-359).
Line 366: “obtained SVPD” should be “obtained with SVPD”
Done (line 380).
Line 383: (Figure 8, referring to VPD)
I think this should be Figure 7B, referring to VPD
Consequently, in that case there is no reference to Figure 8 in the Results section (nor in the rest of the manuscript). Please add a (small) paragraph with the description of the outcomes of the comparison between predicted and observed pheromone release levels (Figure 8).
Correct. We added a sentence in lines 406-411.
- Discussion
Line 437
Additionally, as the temperature of the pheromone increases, its surface tension diminishes, and this facilitates attractive forces between pheromone molecules and other surfaces in the vicinity to pull water into capillary pores of the dispenser wall and wet them out.
This is unclear for me. Are the authors referring to the fact that precipitation and moisture influence the absorption and desorption of water by dispensers? Or what do you mean by “pull water into capillary pores of the dispenser wall and wet them out”?
The sentence was rephrased (lines 468-469).
Line 480
Considering the mechanism of mating disruption of EGVM and EGBM. Is it competitive or non-competitive disruption? Depending on the mechanism what is the most important: the concentration of pheromone at the surface of the dispensers (point sources) (competitive) or the concentration of pheromone in the air of the vineyard (non-competitive: desensitization of males). Using the type of passive dispenser, competitive attraction seems to be the main mechanism acting in disrupted fields (Gavara et al., 2020). I think this is an important element, so can you discuss this and include it in the discussion section?
A paragraph on MD mechanisms was added in lines 510-517.
Line 485: “precipitation and moisture influence the absorption and desorption of water by dispensers.”
However, precipitation had no notable effect on pheromone release in this study. How do you reconcile this with the discussion about “pull water into capillary pores of the dispenser wall and wet them out”? (see line 437 above)
The sentence was rephrased (lines 468-469).

Reviewer 2 Report
Comments and Suggestions for Authors
Fedele Giorgia and colleagues describe in this work equations correlating the effect of weather conditions on pheromone release rate for two commercial dispensers of sex pheromone of Lobesia botrana and Eupoecilia ambiguella. The work analyzes the residual pheromone content of the emitters, justifying it by comparison with previous work. Despite the interesting results, the authors do not explain how these results could actually improve mating disruption treatment with these two commercial dispensers. More importantly, the equations must be validated for this work to be considered an advancement in knowledge of sex pheromone mating-disruption technology for pest control.
Author Response
Fedele Giorgia and colleagues describe in this work equations correlating the effect of weather conditions on pheromone release rate for two commercial dispensers of sex pheromone of Lobesia botrana and Eupoecilia ambiguella. The work analyzes the residual pheromone content of the emitters, justifying it by comparison with previous work. Despite the interesting results, the authors do not explain how these results could actually improve mating disruption treatment with these two commercial dispensers. More importantly, the equations must be validated for this work to be considered an advancement in knowledge of sex pheromone mating-disruption technology for pest control.
Improvement in MD are discussed between lines 535 and 546.
Reviewer 3 Report
Comments and Suggestions for Authors
The paper, entitled "The dynamics of pheromone release in two passive dispensers commonly used for mating disruption in the control of Lobesia botrana and Eupoecilia ambiguella in vineyards," provides valuable insight into the release of pheromone by the dispensers in field conditions. It also offers precise formulas and information about pheromone emission. I would like to congratulate the authors for the excellent work and the high quality of the document. During the review, I only identified two details that need to be adressed:
-L42-44 “Two tortricid moths, the European grapevine moth (EGVM), Lobesia botrana (Denis & Schiffermüller)(Lepidoptera: Tortricidae) and the European grapeberry moth (EGBM), Eupoecilia ambiguella (Hübner)(Lepidoptera: Tortricidae)…”
- The need to increase the quality of the figures, particularly the figure 1.
Author Response
The paper, entitled "The dynamics of pheromone release in two passive dispensers commonly used for mating disruption in the control of Lobesia botrana and Eupoecilia ambiguella in vineyards," provides valuable insight into the release of pheromone by the dispensers in field conditions. It also offers precise formulas and information about pheromone emission. I would like to congratulate the authors for the excellent work and the high quality of the document. During the review, I only identified two details that need to be adressed:
-L42-44 “Two tortricid moths, the European grapevine moth (EGVM), Lobesia botrana (Denis & Schiffermüller)(Lepidoptera: Tortricidae) and the European grapeberry moth (EGBM), Eupoecilia ambiguella (Hübner)(Lepidoptera: Tortricidae)…”
Modified (lines 44-45).
- The need to increase the quality of the figures, particularly the figure 1.
Done.
Round 2
Reviewer 1 Report
Comments and Suggestions for Authors
In the cover letter it is stated that:
"In the discussion, we address the possible role of both rain and RH (lines 488-500)."
? I think this should be lines 468-500.
Line 130. Dispensers were placed in incubators at six constant temperature regimes
Which type of incubator was used? Was the relative humidity controlled (constant) in this incubators?
Information included in the text (line 129).
Only the type of incubators is added. Was the relative humidity controlled (constant) in this incubators?
line 493:
On the other hand, non-competitive mechanism, i.e. the desensitization of males, typically take place when moths are exposed to huge concentration of pheromone, hardly obtained in vineyard treated with MD dispensers [65,66].
I think this should be:
typically takes place when moths are exposed to huge concentration of pheromone, which is hardly obtained in vineyard treated with MD dispensers
Author Response
In the cover letter it is stated that:
"In the discussion, we address the possible role of both rain and RH (lines 488-500)."
? I think this should be lines 468-500.
The potential role of rain and RH is discussed in lines 486-498. In lines 458-485, we discussed the role of VPD and temperature.
Line 130. Dispensers were placed in incubators at six constant temperature regimes
Which type of incubator was used? Was the relative humidity controlled (constant) in this incubators?
Information included in the text (line 129).
Only the type of incubators is added. Was the relative humidity controlled (constant) in this incubators?
No, the relative humidity was not kept constant in the incubators. We measured the relative humidity levels during the experiment and the levels varied among the incubators similarly to the variation observed in field at the same temperature (e.g., at 30°C the RH varied between 39 to 47%).
line 493:
On the other hand, non-competitive mechanism, i.e. the desensitization of males, typically take place when moths are exposed to huge concentration of pheromone, hardly obtained in vineyard treated with MD dispensers [65,66].
I think this should be:
typically takes place when moths are exposed to huge concentration of pheromone, which is hardly obtained in vineyard treated with MD dispensers
Addressed (now lines 511-513).
Reviewer 2 Report
Comments and Suggestions for Authors
The issues addresed in my previous comments have not been improved, particularly in lines 535 and 546 or any other part of the text.
Author Response
The issues addresed in my previous comments have not been improved, particularly in lines 535 and 546 or any other part of the text.
Improvements were discussed in lines 533-551.